# Hot Water Extraction of Antioxidants from Tea Leaves—Optimization of Brewing Conditions for Preparing Antioxidant-Rich Tea Drinks

**DOI:** 10.3390/molecules28073030

**Published:** 2023-03-28

**Authors:** Yan Cheng, Fumin Xue, Yu Yang

**Affiliations:** 1Shandong Analysis and Test Centre, Qilu University of Technology (Shandong Academy of Sciences), Jinan 250014, China; 2Department of Chemistry, East Carolina University, Greenville, NC 27858, USA

**Keywords:** hot water extraction, brewing, tea, polyphenols, caffeine

## Abstract

There are billions of tea drinkers around the world. However, the optimized tea-brewing temperature and time conditions for achieving a higher concentration of antioxidants in tea drinks have not been thoroughly studied. Finding out the optimized brewing conditions can benefit tea drinkers significantly. In this work, we have studied ten antioxidants from seven different popular green, Oolong, black, and scented teas using hot water extraction followed by HPLC analysis. The antioxidant yield was evaluated at 25–100 °C with 5 to 720 min of brewing time. Our results show that the extraction efficiency was enhanced by increasing the water temperature and the highest yield of antioxidants was achieved at 100 °C. The antioxidant yield increased with prolonged brewing time. However, the degradation of antioxidants occurred when tea leaves were extracted for 120 to 720 min. Caffeine was found in all seven tea samples. At 100 °C, the caffein concentration in the tea extract ranged from 7.04 to 20.4 mg/g in Rizhao green tea. Longjing green tea contained the highest concentration of antioxidants (88 mg/g) in the 100 °C extract. Epigallocatechin and caffeine were the most abundant compounds found in all tea samples studied, ranging from 4.77 to 26.88 mg/g. The antioxidant yield was enhanced by increasing the extraction time to up to 60–120 min for all ten compounds studied.

## 1. Introduction

The active ingredients in tea leaves have been widely studied [1,2,3,4,5,6,7,8]. It is well known that tea leaves are rich in antioxidant content [9,10]. The major polyphenol compounds identified in tea leaves include gallic acid, caffeine, (+)-catechin (C), (−)-epicatechin, (−)-gallocatechin, (−)-epigallocatechin, (−)-gallocatechin gallate, (−)-epigallocatechin gallate, and (−)-epicatechin gallate. The contents of each antioxidant typically range 1 to 20 mg/g of tea [3,5,6]. The health benefits of tea drinks have also been discussed [11]. Various extraction techniques have been employed to extract the antioxidants from tea leaves [12,13,14,15,16,17,18,19]. They include subcritical water extraction [16], pressurized hot water extraction (the same as subcritical water extraction) [15], microwave-assisted extraction [19], and cold and hot water extraction [17]. Supercritical fluid extraction using modified carbon dioxide has been used to remove caffeine by retaining catechins in green tea [20]. The decaffeination of fresh green tea leaves by hot water treatment has also been investigated [21].

While the polyphenols in tea leaves have been well studied, as mentioned above, the optimized tea-brewing temperature and time have rarely been studied [12,13,14,15,16,17,18,19]. The polyphenol studies reported in the literature have mostly focused on organic solvent extraction of antioxidants from tea leaves [12,13,14,15,16,17,18,19]. Obviously, the tea extracts obtained by toxic organic solvents cannot be consumed by human tea drinkers. Although the antioxidant contents in water extracts were reported in Reference [3] the tea sample was extracted at only one temperature, 90 °C, and extracted for 30 min only [3]. Therefore, the optimized tea-brewing conditions such as water temperature and extraction time remain unknown. The tea-brewing conditions given in the instructions for tea bags vary widely, from 70 to 90 °C for brewing temperature and 2–10 min for brewing time. Because tea drinkers want to brew antioxidant-rich tea drinks, knowing the optimized tea-brewing temperature and brewing time is of great interest.

Therefore, the objective of this study was to use a large sample size to optimize the tea-brewing conditions. The investigators of this study selected seven most popular green teas, black teas, Oolong teas, and jasmine teas consumed widely in east Asia and around the world to optimize the brewing conditions in order to obtain the richest antioxidant-containing tea drinks.

As we all know, brewing tea is a hot water extraction process. However, what temperature of water should be used and how long should the brewing process last are the questions that many tea drinkers ask. In this work, we have studied ten antioxidants (Appendix A) from seven different green, Oolong, black, and scented tea leaves using hot water extraction followed by HPLC/MS (for characterization and identification) and HPLC/UV (for quantification) separation and analysis. The individual and the total antioxidant yield was evaluated at 25, 50, 70, 80, 90, and 100 °C with 5, 10, 20, 30, 60, 120, and 720 min brewing times.

## 2. Results and Discussion

### 2.1. HPLC Method Development

The HPLC quantification method was developed first. Table 1 shows the regression equation, correlation coefficient, and limit of detection of each antioxidant. The correlation coefficient ranged from 0.9988 to 1.000. The limit of detection was between 0.002 and 0.020 mg.

Under optimized conditions (see Section 3.4 and Figure 1 legend), all ten antioxidants in both standard and sample solutions were well separated from all seven different tea samples, as shown in Figure 1. LC/MS analysis confirmed that each antioxidant peak had no coelutions. Thus, the separation efficiency was adequate.

### 2.2. Temperature Effect on Extraction Efficiency

Appendix A and Figure 2 show a clear temperature effect on the extraction efficiency of the antioxidants. The quantity of antioxidants extracted increased with increasing temperature. Although the yield obtained at 90 °C was almost the same as that achieved at 100 °C, the quantity of antioxidants extracted from all seven types of tea leaves at 100 °C was always higher than that at 90 °C.

As shown in Appendix A and Figure 3, caffeine was found in all seven tea leaves. At 100 °C, the caffeine concentration in the tea extract ranged from 7.04 mg/g in Da Hong Pao Oolong tea to 20.4 mg/g in Rizhao green tea. Longjing green tea contained the highest concentration of antioxidants in the 100 °C extract, about 68 mg of polyphenols per gram of tea and approximately 20 mg of caffeine per gram of tea leaves; a total of 88 mg of antioxidants in each gram of tea leaves. This was followed by Rizhao green tea with approximately 81 mg antioxidants per gram of tea. Only gallic acid and caffeine were found in Pu-erh tea. Among all the tea leaves studied, Pu-erh tea contained the lowest total concentration of antioxidants, about 8 mg/g, as shown in Appendix A and Figure 3.

Epigallocatechin and caffeine had the highest concentrations in the tea drinks obtained from all tea leaves studied, as shown in Appendix A, ranging from 4.77 to 26.88 mg/g for epigallocatechin and 7.04 to 20.44 mg/g for caffeine at 100 °C. On the other hand, gallocatechin gallate and catechin gallate yielded the lowest quantity with a range of 0.14 to 0.92 mg/g for gallocatechin gallate and 0.14–0.90 mg/g for catechin gallate at 100 °C. As a matter of fact, gallocatechin gallate was not detectable in Pu-erh tea and Zheng Shan Xiao Zhong tea at any temperatures and in Da Hong Pao tea and Jasmine tea at temperatures lower than 80 °C as shown in Appendix A. Similarly, catechin gallate was not detected in Pu-erh tea, Zheng Shan Xiao Zhong, and Jasmine tea leaves at any of the temperatures employed.

### 2.3. Effect of Extraction Time on Extraction Efficiency

The effect of extraction time on the extraction yield was evaluated at 100 °C. As shown in Figure 4 and Appendix A, the antioxidant yield was enhanced by increasing extraction time to up to 120 min for all ten compounds studied. Except GA, EGC, and CAF, the other seven antioxidants started degrading after 120 min, and this is clearly seen in Appendix A.

### 2.4. Discussion

The quality of this method is comparable to that of similar HPLC methods developed by other researchers [3,5,6]. Each analyte’s peak was confirmed by LC/MS analysis and there was no co-elution for any of the ten antioxidants investigated. The chemical compounds found in tea leaves in this study agree well with those reported in the literature [3,5,6,22,23]. The vast majority of the studies reported in the literature involve the use of organic solvents to extract antioxidants from tea leaves. These organic solvent extraction techniques used on tea leaves are not tea-brewing processes. The objective of these studies in the literature was to evaluate the antioxidant contents in tea leaves, not for preparing antioxidant-rich tea drinks because the toxic tea extracts cannot be consumed by humans [5,6,22,23]. It must be pointed out that the hot water extraction used in this study is a simulation method for the tea drink brewing process. The water extracts obtained in this study are tea drinks for human consumption. The content of antioxidants from the tea leaves in the water extracts are also similar to the literature findings [3]. For example, the antioxidant contents in the extracts obtained after the extraction in 90 °C water typically ranged from 0.5 to 30 mg/g of tea [3], which is similar to those obtained in our study, as shown in Appendix A.

As shown in Figure 2 and Appendix A, the antioxidant concentration in the tea drinks (water extracts) increased with water temperature. This makes sense since the polarity of water decreases with increasing the water temperature, making higher-temperature water more compatible for removing antioxidants from tea leaves into the water. Figure 4 and Appendix A demonstrate the effect of extraction time on antioxidant contents. The concentration of antioxidants increased with a prolonged extraction time of up to 120 min. However, from 120 to 720 min, the concentration of analytes in the tea drinks decreased. This might be caused by degradation of the antioxidant compounds.

## 3. Experimental Section

### 3.1. Reagents and Materials

Gallic acid (GA), (+)-gallocatechin (GC), (−)-epigallocatechin (EGC), cianidanol (C), L-epicatechin (EC), (−)-epigallocatechin gallate (EGCG), (−)-gallocatechin gallate (GCG), (−)-epicatechin gallate (ECG), catechin gallate (CG), caffeine (CAF) were all purchased from Shanghai National Pharmaceutical Company (Shanghai, China). HPLC grade acetonitrile was obtained from Tedia Company Inc. (Fairfield, OH, USA). 18.2 MΩ-cm (25 °C) deionized water was prepared in our lab using a Sartorius Lab Gmbh & Co. KG arium mini system (Goettingen, Germany). As described in Table 2, seven different tea samples from four different classes (green tea, Oolong tea, black tea, and scented tea) were purchased from local tea shops. A H-Class UHPLC system (Waters, Milford, MA, USA) combined with a Q-TOF mass spectrometer equipped with an ESI interface (Impact II, Bruker, Germany) were employed in the qualitative analysis of antioxidants. A model 1260 HPLC system with G7114A 1260 VWD detector (Serial No. DEACX12311) and G7129A 1260 Vial sampler (Serial No. DEAEQ24624) and G7111B 1260 Quat Pump (Serial No. DEAEW04667) was acquired from Agilent Technologies Co. Ltd. (model 1260, Santa Clara, CA, USA) and used for quantitative analysis. An InertSustain C18 column (5 μm, 4.6 × 50 mm) was purchased from Shimadzu Corporation (Kyoto, Japan).

### 3.2. Hot Water Extraction

Approximately 0.2000 to 0.2200 g of tea leaves was accurately weighed and placed inside a 25 mL beaker. Then 10.00 mL of hot water (at 100, 90, 80, 70, 50, or 25 °C) was added to the beaker and the beaker was covered using a watch glass. Triplicate experiments were carried out. After 20 min of static water extraction (the beakers were exposed to room temperature), the water extractant was filtrated using 0.45 μm filter paper and collected in a glass vial for HPLC analysis.

The experiments of evaluating the effect of extraction time on extraction efficiency were conducted at 100 °C. Approximately 0.2000 to 0.2200 g of tea leaves was accurately weighed and placed inside a 25 mL beaker. Then 10.00 mL of 100 °C water was added to the beaker and the beaker was covered with a watch glass. Then the beaker was placed at room temperature without heating. Please note that experiments for each different extraction time were conducted individually. Each experiment of a different extraction time lasted from 0 min to the time as reported in Appendix A and Figure 4. The extraction times investigated were 5, 10, 20, 30, 60, 120, and 720 min. After each extraction, the water extractant was filtrated using 0.45 um filter paper and collected in a glass vial for HPLC analysis.

### 3.3. LC/MS Qualitative Analysis

The LC/MS analysis was performed by a H-Class UHPLC system combined with a Q-TOF mass spectrometer equipped with an ESI interface. The *m*/*z* range was from 50 to 1200. The capillary voltage was 3500 V in the positive mode and 3000 V in the negative mode. The nebulizer pressure was 2.0 bar, and the flow rate of dry gas was 8.0 L/min. The drying gas temperature was 200 °C.

### 3.4. HPLC Quantitative Analysis

Stock solutions were prepared by weighing 10.0 mg of gallic acid (GA), (+)-gallocatechin (GC), (−)-epigallocatechin (EGC), cianidanol (C), L-epicatechin (EC), (−)-epigallocatechin gallate (EGCG), (−)-gallocatechin gallate (GCG), (−)-epicatechin gallate (ECG), catechin gallate (CG), caffeine (CAF). Each antioxidant was then placed separately into a 50.00 mL volumetric flask and deionized water was added to each volumetric flask to the mark. HPLC working standard solutions were prepared using the above-mentioned stock solutions.

A 1260 Agilent HPLC system was employed. The separation was achieved on a GL Sciences column (InertSustain C18, 5 µm, 4.6 × 250 mm). Mobile phase A contained 90 mL of acetonitrile, 20 mL of acetic acid, and 2 mL of EDTA in 1 L solution. Mobile phase B contained 800 mL of acetonitrile, 20 mL of acetic acid, and 2 mL of EDTA in 1 L solution. The gradient condition is shown in Table 3. The mobile phase flow rate was 1.0 mL/min, and the separation column was maintained at 35 °C. UV detection was set at 278 nm.

Once the mobile phase back pressure and the column oven temperature were stabilized, blank runs were conducted to ensure that the HPLC was ready to go for analysis. Then 20 μL of working solution or sample solution was injected into the HPLC system.

## 4. Conclusions

An HPLC method was successfully developed for the separation and analysis of ten antioxidants contained in tea leaves. HPLC/MS analysis confirmed that there was no coelution of the antioxidant peaks. Therefore, the separation efficiency was good. The correlation coefficient for the calibration curves ranged from 0.9988 to 1.000. The limit of detection was between 0.002 and 0.020 mg. The antioxidant extraction efficiency was enhanced by increasing the water temperature. The highest yield of antioxidants was achieved at 100 °C. The total antioxidant content was enhanced 3–10 fold by increasing the extraction temperature from 25 to 100 °C. For all seven tea leaves investigated, caffeine had the highest concentration in the tea drinks prepared by hot water extractions. The concentration of caffeine in the water extract typically ranged from 10 to 20 mg/g. The highest total concentration of antioxidants in the water extracts was found in Longjing green tea. The total antioxidant concentration in the water extracts increased with a longer extraction time. However, the total antioxidant content in the water extracts decreased when the extracted time was increased from 120 to 720 min, indicating that antioxidants might undergo decomposition during the prolonged extraction time. In closing, this study found that the optimized hot water extraction temperature and time for obtaining antioxidant-rich tea drinks were 100 °C and 120 min.

## Figures and Tables

**Figure 1 molecules-28-03030-f001:**
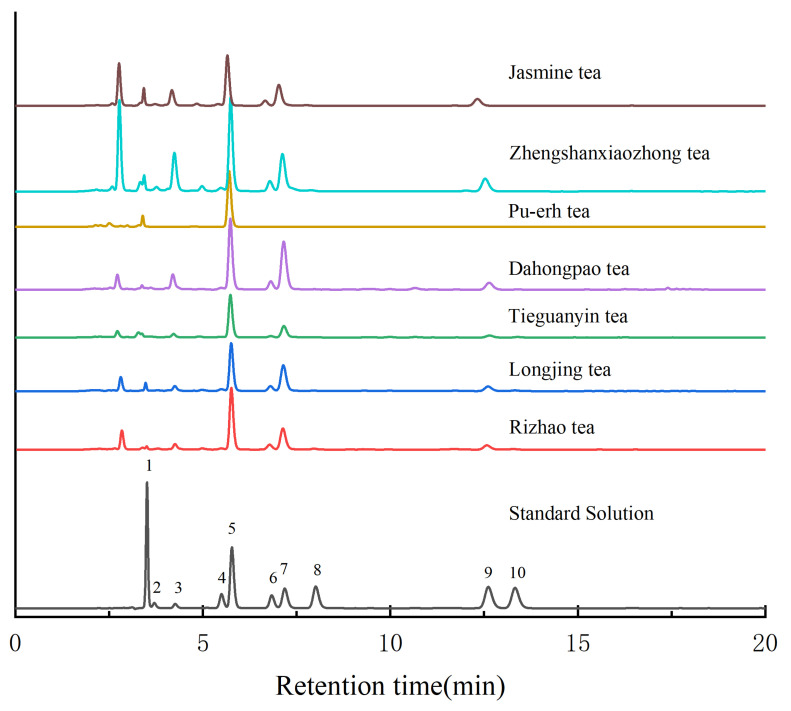
HPLC chromatograms of seven tea samples and standard solution at acquisition wavelength of 278 nm. The mobile phase flow rate was 1.0 mL/min, and the separation column was maintained at 35 °C. The gradient elution conditions are given in Section 3.4. Peak identification: 1: gallic acid; 2: gallocatechin; 3: epigallocatechin; 4: cianidanol; 5: caffeine; 6: L-epicatechin; 7: epigallocatechin gallate; 8: gallocatechin gallate; 9: epicatechin gallate; 10: catechin gallate.

**Figure 2 molecules-28-03030-f002:**
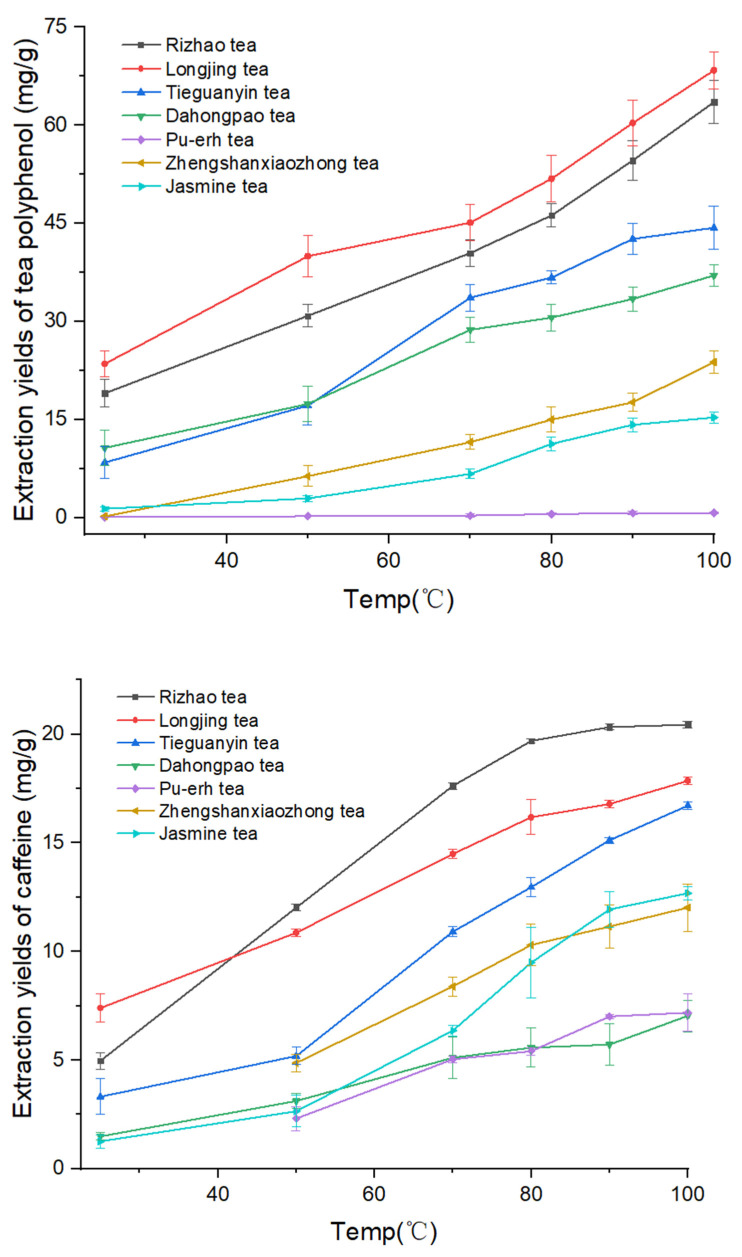
Effect of brewing temperature on the yield of polyphenols (**top**) and caffeine (**bottom**).

**Figure 3 molecules-28-03030-f003:**
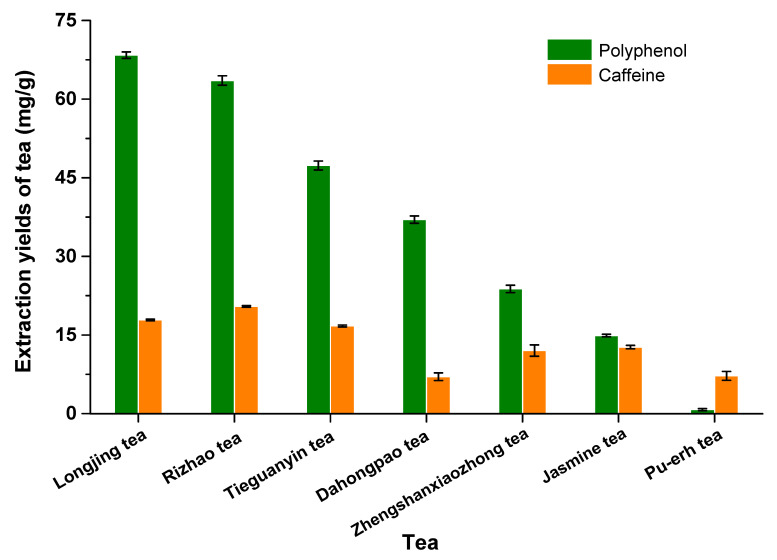
Extraction yields of polyphenols and caffeine from seven tea samples extracted with water at 100 °C for 20 min.

**Figure 4 molecules-28-03030-f004:**
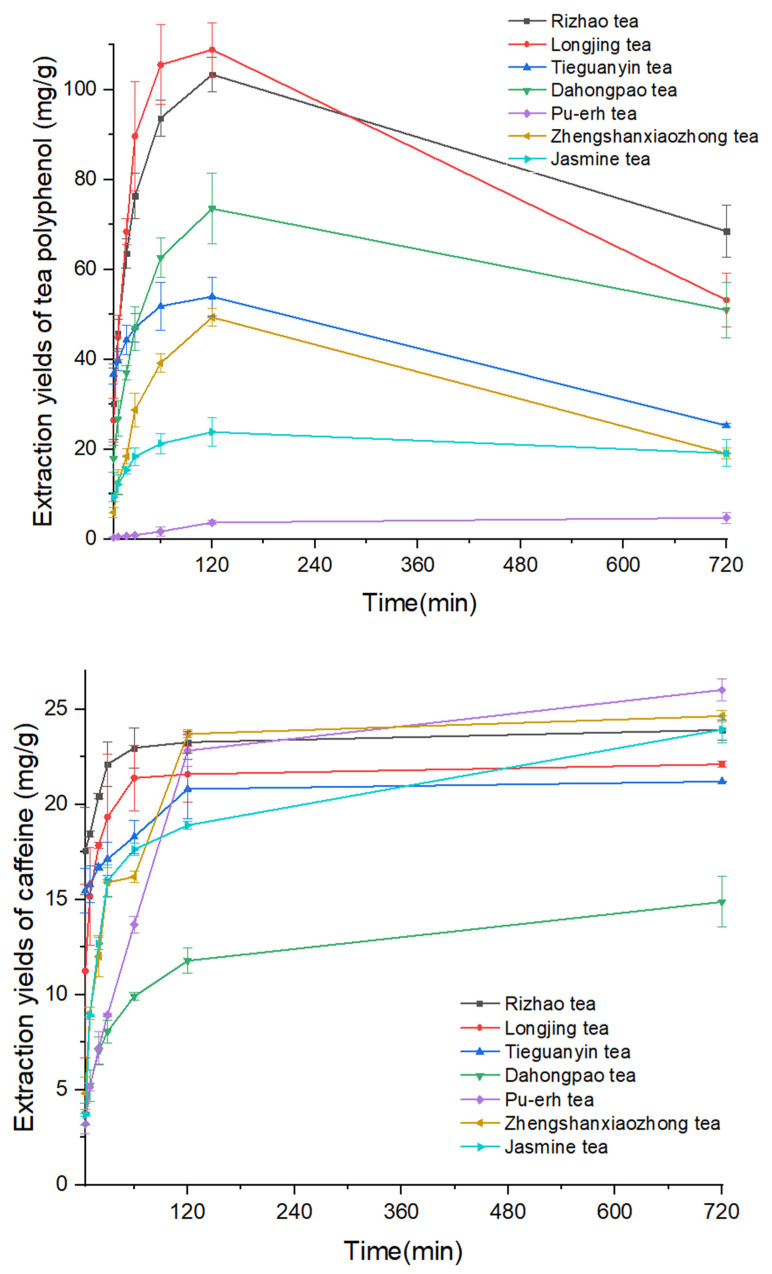
Effect of brewing time on the yield of polyphenols (**top**) and caffeine (**bottom**).

**Table 1 molecules-28-03030-t001:** Regression equation, correlation coefficient, and limit of detection.

Analyte	Regression Equation	R²	LOD * mg
GA	y = 3131.1x + 9.3761	0.9999	0.002
GC	y = 259.05x + 1.6035	0.999	0.020
EGC	y = 171.48x + 0.6164	0.999	0.020
C	y = 682.55x + 0.4279	0.999	0.010
CAF	y = 2775.9x + 2.7423	0.999	0.002
EC	y = 742.64x + 0.7184	0.999	0.010
EGCG	y = 1104.7x + 0.8841	0.999	0.010
GCG	y = 1404.1x − 0.3721	0.999	0.010
ECG	y = 1827x − 0.4005	0.999	0.005
CG	y = 1799.4x − 8.9632	0.999	0.005

* LOD = 3S (standard deviation of the blank)/m (slope of the calibration curve) based on triplicate measurements.

**Table 2 molecules-28-03030-t002:** Descriptions of tea leaves studied.

Type	Name	Place	Coordinates
Green tea (no fermentation)	Rizhao	Bojiakou, Shangdong Province, China	35.29, 119.25
Green tea (no fermentation)	Longjing	Hangzhou, Zhejiang Province, China	30.15, 120.20
Oolong tea (half fermentation)	Tieguanyin	Anxi, Fujian Province, China	25.07, 118.18
Oolong tea (half fermentation)	Dahongpao	Mountain Wuyi, Fujian Province, China	27.67, 117.97
Black tea (full fermentation)	Pu-erh	Menghai, Yunnan Province, China	21.96, 100.45
Black tea (full fermentation)	Zhengshanxiao- zhong	Mountain Wuyi, Fujian Province, China	27.42, 117.39
Scented tea	Jasmine	Yuanjiang, Yunnan Province, China	23.61, 102.03

**Table 3 molecules-28-03030-t003:** HPLC gradient conditions.

Time (min)	Mobile Phase A
0	90%
20.0	85%
30.0	60%
30.1	90%
40.0	90%

## Data Availability

Not applicable.

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
