# Peer review of "Hot Water Extraction of Antioxidants from Tea Leaves—Optimization of Brewing Conditions for Preparing Antioxidant-Rich Tea Drinks"

_molecules, 2023, doi:10.3390/molecules28073030_

Round 1

Reviewer 1 Report

I went through your manuscript entitled "Hot Water Extraction of Antioxidants from Tea Leaves – Optimization of Brewing Conditions for Preparing Antioxidant-Rich Tea Drinks". Through there are some important findings are there, overall the manuscript is not up to the level of Molecules. I have given my suggestions below.

1. Authors must mention the novelty aspects of the study in abstract (briefly) and introduction (in details). Since, Tea polyphenols are well studied, readers won't find the novelty of the work easily.

2. A clear-cut information on the chemistry (both qualitative and quantitative) of Tea must be made in the introduction

3. Regarding the figure 1, most of the readers will be familiar with the structures of these polyphenolics.. so it is not a main finding and can be moved as supplemental information

4. If the authors can provide precise location in table 1 as coordinates it will be more accurate 

5. HPLC alone won't be enough to say the peaks in Tea samples are formed exclusively of the standard compounds tested. I recommend compounds analysis using LCMS during revision. Or else authors need to justify the findjng using previously published data (with exact same chromatography conditions).

6. Figure 3 lack standard deviation or standard error information 

7. Why the authors choose hot water extraction? Whether is it's significance 

8. Combining results with discussion is not an easy way for presentation. Authors need to take extra efforts to improve the discussion portions. Or else make it a stand alone section during revision that will help authors to improve in depth analysis

Author Response

I went through your manuscript entitled "Hot Water Extraction of Antioxidants from Tea Leaves – Optimization of Brewing Conditions for Preparing Antioxidant-Rich Tea Drinks". Through there are some important findings are there, overall the manuscript is not up to the level of Molecules. I have given my suggestions below.

  1. Authors must mention the novelty aspects of the study in abstract (briefly) and introduction (in details). Since, Tea polyphenols are well studied, readers won't find the novelty of the work easily.

The novelty of this work was addressed in the Abstract and Introduction of our original manuscript (ms). The novelty of this study was also reflected in the title “….Optimization of Brewing Conditions for Preparing Antioxidant-Rich Tea Drinks.” There are billions of tea drinkers all around the world, especially in east Asia. However, the optimized brewing temperature and time have not been studied. Brewing instructions on tea bags normally vary from 70 to 90 ℃ for brewing temperature and 2-10 min for brewing time.  Therefore, using large sample size (seven most popular green teas, black teas, Oolong teas, and jasmine tea widely consumed in Asia and around the world) to optimize the brewing conditions in order to obtain the richest antioxidant containing tea drinks benefits billions of tea drinkers. This is the novelty of this study. This point was elaborated in the revised ms.  In addition, the vast majority of the published tea polyphenol studies concentrated on organic solvent extractions, characterization, and quantifications of polyphenols. Our work is the first tea brewing simulation study with large sample size (seven different teas). Therefore, the novelty of this study is significant. And especially all tea drinkers will benefit greatly from the findings in this work. This point was made clearer in the revised ms.

  1. A clear-cut information on the chemistry (both qualitative and quantitative) of Tea must be made in the introduction

More information on the chemistry of tea was added to the revised ms.

  1. Regarding the figure 1, most of the readers will be familiar with the structures of these polyphenolics.. so it is not a main finding and can be moved as supplemental information

We agree with the reviewer. Figure 1 was presented as supplemental materials in the revised ms.

  1. If the authors can provide precise location in table 1 as coordinates it will be more accurate 

This is a great suggestion. The coordinates were added in the revised ms as requested.

  1. HPLC alone won't be enough to say the peaks in Tea samples are formed exclusively of the standard compounds tested. I recommend compounds analysis using LCMS during revision. Or else authors need to justify the findjng using previously published data (with exact same chromatography conditions).

The identification of each antioxidant peak was confirmed by LC/MS. This information was added to the revised ms.

  1. Figure 3 lack standard deviation or standard error information 

Fixed

  1. Why the authors choose hot water extraction? Whether is it's significance 

As described in the original ms, the objective of this study was to optimize the tea brewing conditions in order to obtain antioxidant-rich tea drinks. Because tea brewing process is a hot water extraction, then hot water extraction has to be employed in this study.

  1. Combining results with discussion is not an easy way for presentation. Authors need to take extra efforts to improve the discussion portions. Or else make it a stand alone section during revision that will help authors to improve in depth analysis

Extensive discussions were added to the revised ms.

Reviewer 2 Report

Abstract is too long – according to the Instruction for Authors: “The abstract should be a total of about 200 words maximum.

Line 57, 59 and others: In scientific papers, we usually don't use first or third person singular narration. If really necessary, please use the wording: Authors...

Introduction does not contain information on the current knowledge and research on the research topic described by the Authors.

The placement of Table 1 at the end of the introduction section is unfortunate.

The Authors did not confirm the results of the qualitative analysis in their work. If you have a DAD detector (most likely, because the Authors have provided only laconic information about the HPLC system used) - you can easily use the peak purity function or check the similarity of the obtained UV spectra of analytes from tea samples with standard spectra. This makes it possible to detect possible coelution. Without such confirmed results of the qualitative analysis the quantitative analysis may lead to erroneous conclusions. Therefore, I would recommend supplementing the work with such measurements to meet common practice and to make the scientific value of this work acceptable.

Line 112 - when writing that a compound is "not detectable", the limit of detection should be given.

The Authors also did not present other - required in such a case - parameters characterizing the applied determination method (i.e. validation parameters), such as equations of calibration curves and others.

I have doubts whether the data in the tables (Table 2 and 3) are a good way to present the results obtained. I think line or bar charts would allow them to be interpreted more easily. In addition, some of the results could be placed in supplementary materials.

The work lacks sufficient data discussion - comparisons of the results obtained with the works of other authors, references to regulations and to recommended doses of antioxidants, etc.

Line 142: model 1260 - please provide full specification of HPLC system used.

The work should be reviewed very carefully in terms of the correctness of the English language and the style of writing a scientific article. Some examples: line 41: "the chemical composition in tea leaves" - is incomprehensible; line 149 "to the beaker and the beaker" - repetition; line 153: "kinetic extractions" - this term is incorrect, and unclear... etc

Lines 187-192 sound like an article from a popular science magazine

Lines 193-208 should certainly not be part of the conclusions

Author Response

Reviewer 2

Abstract is too long – according to the Instruction for Authors: “The abstract should be a total of about 200 words maximum.”

The abstract was reduced to 203 words.

Line 57, 59 and others: In scientific papers, we usually don't use first or third person singular narration. If really necessary, please use the wording: Authors...

Fixed

Introduction does not contain information on the current knowledge and research on the research topic described by the Authors.

The requested information was added to the revised ms.

The placement of Table 1 at the end of the introduction section is unfortunate.

This table was moved up in the revised ms.

The Authors did not confirm the results of the qualitative analysis in their work. If you have a DAD detector (most likely, because the Authors have provided only laconic information about the HPLC system used) - you can easily use the peak purity function or check the similarity of the obtained UV spectra of analytes from tea samples with standard spectra. This makes it possible to detect possible coelution. Without such confirmed results of the qualitative analysis the quantitative analysis may lead to erroneous conclusions. Therefore, I would recommend supplementing the work with such measurements to meet common practice and to make the scientific value of this work acceptable.

LC/MS experiments confirmed that there were no coelutions of antioxidant peaks in the HPLC chromatograms. This information was added to the revised ms.

Line 112 - when writing that a compound is "not detectable", the limit of detection should be given.

LOD was given in the revised ms.

The Authors also did not present other - required in such a case - parameters characterizing the applied determination method (i.e. validation parameters), such as equations of calibration curves and others.

More details regarding the method were added to the revised ms.

I have doubts whether the data in the tables (Table 2 and 3) are a good way to present the results obtained. I think line or bar charts would allow them to be interpreted more easily. In addition, some of the results could be placed in supplementary materials.

As requested, charts were created and used to present the effects of brewing temperature and brewing time on extraction efficiency, and this was added to the revised ms. We presented Tables 2 & 3 as supplemental materials in the revised ms.

The work lacks sufficient data discussion - comparisons of the results obtained with the works of other authors, references to regulations and to recommended doses of antioxidants, etc.

More in-depth discussion was added to the revised ms.

Line 142: model 1260 - please provide full specification of HPLC system used.

Revised as requested.

The work should be reviewed very carefully in terms of the correctness of the English language and the style of writing a scientific article. Some examples: line 41: "the chemical composition in tea leaves" - is incomprehensible; line 149 "to the beaker and the beaker" - repetition; line 153: "kinetic extractions" - this term is incorrect, and unclear... etc

Thorough English editing was conducted and the entire ms was revised to correct any incomprehension and unclearness.

Lines 187-192 sound like an article from a popular science magazine

This paragraph was deleted and replaced with new text in the revised ms.

Lines 193-208 should certainly not be part of the conclusions

Summary of results should be provided in conclusions. However, we have significantly revised this paragraph.

Round 2

Reviewer 1 Report

No more comments

Author Response

N/A

Reviewer 2 Report

The Authors have significantly improved the work by supplementing it with additional information and improving the English language. Nevertheless, the manuscript still requires additional modifications.

Line 11: "optimized tea brewing temperature (...)" - i.e. optimized towards what?

Line 43: "tea brewing temperature and time have rarely been studied" - i.e. please give some references and discuss very shortly the results from other papers. Against this background, it is possible to present what the Authors have done in this work. Otherwise it is not clear what is the novelty of your work.

Table 1 - should be included in the experimental part.

Line 75: “We first developed” – this sentence would sound better in the passive form.

Table 2:

1) there is no definition given when determining the LOD and LOQ.

2) R2=1.000 - in fact it is true for a line passing through two points. In my opinion, in the case of experimental data, when measurement points are subject to errors, this value should be 0.999.

3) What was the number of repetitions (n)?

Line 83: "Under optimized conditions" - what was optimized? I guess - HPLC separation parameters. If so, it would be appropriate to describe the experiments related to this optimization. Otherwise, do not use the word "optimized".

Line 111 and others: “most abundant compounds” – I understand the substances with the highest concentration in a sample – the word “abundant” is a jargon abbreviation.

The paper also lacks a discussion related to the comparison of the Authors' results with other papers. In lines 137 - 138 there is only a laconic mention of this - it is far too little. Without reference to the literature (in the introduction, but also in the discussion of the results), the work may seem not very revealing - please prove the novelty of your work by referring to the current state of knowledge.

Line 177 and others: "Kinetic extractions" - is a jargon abbreviation - I understand that it is an extraction carried out in dynamic conditions, where time is a variable, and samples were taken at intervals. Does Figure 4 refer to this term? This experiment has not been described in detail. I understand that the teas were brewed with water at a certain initial temperature, and then they just cooled down? The experimental part, in my opinion, still contains ambiguities and should be described more precisely.

Section Conclusions has been corrected, but lines 223-228 have simply been pasted from the experimental section  (lines 102-108), similarly 229-233 is a copy of lines 111-115. “Conclusions” are not “copy and paste” results. Please draw CONCLUSIONS, not copy the text.
